# Commute Your Domains: Trajectory Optimality Criterion for Multi-Domain Learning

## Abstract

In multi-domain learning, a single model is trained on diverse data domains to leverage shared knowledge and improve generalization. The order in which the data from these domains is used for training can significantly affect the model's performance on each domain. However, this dependence is under-studied. In this paper, we investigate the influence of training order (or data mixing) in multi-domain learning using the concept of Lie bracket of gradient vector fields. By analyzing the infinitesimal effects of changing the training order, we identify regions in the parameter space where altering the order between two training domains can benefit the target loss. We validate the predictions of our theoretical framework on the influence of training order (or data mixing) both on a toy example and bilingual LLM pre-training.

## 1 Introduction

In real-world scenarios, training data may come from different sources that vary in quality, topics, diversity, and other aspects. For example, the modern large language models are trained on data collected from the curated list of domains that are comprised of web-crawled data, math, code, academic papers, etc. The natural question arises: how to mix the data and when to use each of the domains (early in the training or in the last stage)? There is no rigorous approach to this problem and practitioners generally use handcrafted solutions.

One prominent example is the multilingual setting, where the data comes from different languages that can also substantially differ in the amounts of available data. It has been demonstrated in (Choi et al., 2024) that the sequence of domain exposure matters in the presence of dataset imbalances. Specifically, training first with the prevalence of high-resource domains followed by an equal mix of high- and low-resource domains yields better results. Motivated by these findings, we address the problem of the domain order during the training for a fixed training budget.

Another relevant directions to our setting are multi-task training, domain adaption and curriculum learning. For multi-task problems, we should balance the signals from different losses/problems to get an optimal training trajectory. There is a big amount of literature on multi-task learning, we mention just a few (Navon et al., 2022; Lee et al., 2022; Yu et al., 2020; Wang et al., 2021). Many of them manipulate with the gradients to stir the training into non-conflicting direction. Different from this papers, we do not interfere gradients of domains as we only concerned with the their order. Domain adaptation concerned with the gradual distribution shift from the source to the target domain (e.g. see (Kumar et al., 2020; He et al., 2023; Zhuang et al., 2024)). The order of the intermediate or mixtures of the source and target domains have a great impact on the adaptation process. In the present paper, we do not interpolate between domains, but, only consider interaction between them to improve training trajectory.

More formally, we consider a scenario where the total amount of data for each domain is fixed, but the order in which the examples from different domains are interleaved can be adjusted. This setup is practical as, for example, we have vast amounts of web-crawled data and limited amounts of high-quality data that we want to fully embrace. If we use the same proportion of high-quality data in each batch it may lead to sub-optimal training and a common practice is to increase the proportion of that data later in the training.

Our key contributions are as follows:

- We introduce a theoretical framework to predict how changes in training order affect model performance.
- Our method provides guidance on how to adjust domain-specific training weights at each step of a training trajectory to achieve better model performance.
- We validate the predictions of our theoretical framework both on a toy example and bilingual LLM pre-training.

## 2 MULTI-DOMAIN LEARNING

In multi-domain learning, where models are trained on data from multiple sources, the choice of domain weights plays a crucial role in balancing the contribution of each domain during training. The weights determine the importance of each domain and influence the model's generalization capabilities.

Particularly, given $K$ supervised datasets with inputs $(X_k)_k$ and outputs $(Y_k)_k$, the target is to find model parameters $\theta^*$ which minimize datasets' losses $(L_1(X_1, Y_1, \theta), \ldots, L_K(X_K, Y_K, \theta))$. The common setups here include multi-objective optimization, in which one deals with the entire Pareto front simultaneously, or loss averaging scenario. In the last one, the target is the weighted sum of the datasets' losses. The training process in this case is based on the reweighting of domain gradients:

$$\theta_{i+1} = \theta_i - \eta \sum_k w_k^i \mathbb{E}_{(x_k, y_k)} \nabla_{\theta_i} L_k(x_k, y_k, \theta_i) \tag{1}$$

where $(x_k, y_k)$ is sampled from $k$-th dataset $(X_k, Y_k)$ and $w_k^i$ are the domain weights at time step $i$. We can equivalently re-phrase (1) as sampling data from the domains into a batch with probabilities $w_k^i$ (we set a constraint $\sum_k w_k^i = 1$) at step $i$. In practice, the usage of this formulation leads to a better performance. Thus, we will follow it in our experiments.

## 3 VECTOR FIELDS AND THEIR COMMUTATORS

We assume that there are $K$ domains, which define a tuple of loss functions $(L_k)_k$ on the space of parameters $\Theta$. Note that we consider the loss functions as something aggregated over the whole dataset, rather than something depending on a training sample $L' = L'(\theta, x, y)$. This space of parameters is considered to be an open subset of $\mathbb{R}^n$. The functions $L_k$ are assumed to be $C^2$-smooth, and therefore have gradients $(\nabla L_k)_k$, forming *vector fields* on $\Theta$.

**Remark.** *From the differential geometry viewpoint, the gradient is well-defined only in case of prescribed Riemann metric on $\Theta$, which in this paper we assume to be identity in standard Euclidean coordinates on $\mathbb{R}^n$, which means that $\nabla f = (\frac{\partial}{\partial x_i} f)_i$. It worth mention that though it is a standard assumption, this choice of Riemann metric is not canonical. Also, it does not resemble all modern training tricks. For example, variable learning rate should be considered as time-dependent scalar Riemann metric; coordinate-wise gradient scaling in modern optimizers, like Adam (Kingma & Ba, 2014) or AdamW (Loshchilov & Hutter, 2017), – as time-dependent Riemann metric with diagonal matrix. Thorough theoretical analysis of these cases is beyond the scope of this paper.*

Training procedure for multi-domain learning requires at each step the choice of weights for each domain, which is generally implemented as composing a batch from samples from different tasks. Given the domain weights as function $w(t) = (w_k(t))_k$ (we call it *weight schedule*) which we require to be non-negative, right-continuous and satisfying $\sum_k w_k(t) = 1$, at time $t$ our loss would be the weighted sum of domain losses $L_k$ with weights $w_k(t)$. Therefore, the parameters $\theta(t)$ satisfy an ODE

$$\dot{\theta}(t) = -\sum_k w_k(t) \nabla L_k(\theta(t)). \tag{2}$$

Recall also the definition of *flow* of a vector field $v$, that is a map $\Phi : \mathbb{R}_+ \times \Theta \to \Theta$ satisfying

$$\dot{\Phi}(t, \theta) = v(\Phi(t, \theta)) \tag{3}$$

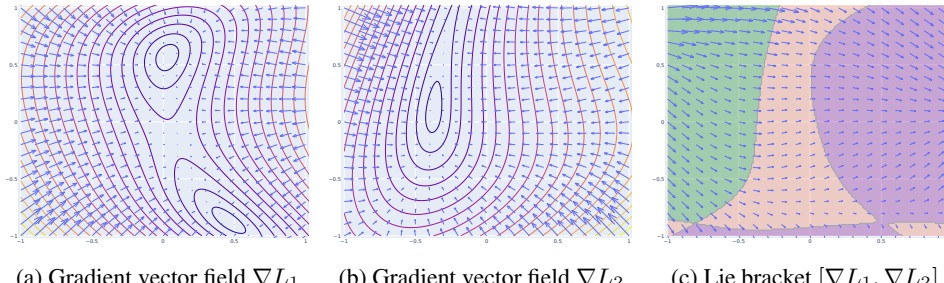

(a) Gradient vector field $\nabla L_1$     (b) Gradient vector field $\nabla L_2$     (c) Lie bracket $[\nabla L_1, \nabla L_2]$

Figure 1: Example of vector fields Lie bracket. (a), (b): the level curves and gradients for two functions $L_1, L_2$. Vectors in (c) correspond to Lie bracket of gradient vector fields. In the green area, the Lie bracket has positive dot products with both $\nabla L_1$ and $\nabla L_2$, and in purple area similar dot products are negative. It means (see Corollary 3.2) that in these areas some reordering of the default training would benefit both losses.

and say that flows $\Phi_1, \Phi_2$ *commute* if for any $t_1, t_2, \theta$ holds

$$(\Phi_1(t_1) \circ \Phi_2(t_2))(\theta) = (\Phi_2(t_2) \circ \Phi_1(t_1))(\theta).$$

If all flows for the vector fields $(-\nabla L_k)_k$ commute (that is, each pair of them commutes), then the result of training using ODE (2) would only depend *on the amount of used training data* $\left( \int w_k(t) \mathrm{d}t \right)_k$ in each domain, rather than on the entire weight schedule $(w_k)_k$. In other words, the training order would not matter at all, for example, one would train on whole domains one by one in any order.

However, this doesn't hold in practice as evidenced by the catastrophic forgetting. If we continue training a model on a different task, its performance on the initial task generally degrades. Hence, we should expect that the vector fields $\nabla L_k$ do not commute, and the training results depend on the order or sampling proportion schedule. In Figure 1, we present such a situation.

### 3.1 TOY EXAMPLE

Suppose we have two domains, training loss functions of which are quadratic:

$$L_{1,2}(\theta) = \frac{1}{2}(\theta - b_{1,2})^T A_{1,2}(\theta - b_{1,2})$$

where $A_{1,2}$ are positive-definite matrices.

The learning on these domains defines vector fields $v_\alpha(\theta) = A_\alpha(\theta - b_\alpha)$, $\alpha = 1, 2$ and therefore flows $\Phi_\alpha(t, \theta) = b_\alpha + e^{tA_\alpha}(\theta - b_\alpha)$. The composition of flow $\Phi_1$ with time $t_1$ and flow $\Phi_2$ with time $t_2$ in different orders yield different results:

$$(\Phi_1(t_1) \circ \Phi_2(t_2))(\theta) = b_1 + e^{t_1 A_1}(b_2 + e^{t_2 A_2}(\theta - b_2) - b_1) =$$
$$= e^{t_1 A_1} e^{t_2 A_2} \theta + (1 - e^{t_1 A_1})b_1 + e^{t_1 A_1}(1 - e^{t_2 A_2})b_2$$
$$(\Phi_2(t_2) \circ \Phi_1(t_1))(\theta) = b_2 + e^{t_2 A_2}(b_1 + e^{t_1 A_1}(\theta - b_1) - b_2) =$$
$$= e^{t_2 A_2} e^{t_1 A_1} \theta + (1 - e^{t_2 A_2})b_2 + e^{t_2 A_2}(1 - e^{t_1 A_1})b_1$$

Note that when matrices $A_1$ and $A_2$ do not commute, $e^{t_2 A_2} e^{t_1 A_1}$ does not equal $e^{t_1 A_1} e^{t_2 A_2}$, which yields giving different dependency of the above formulae on $\theta$. Moreover, even if $A_1$ and $A_2$ commute, the result of flow composition differ by a constant vector $(1 - e^{t_1 A_1})(1 - e^{t_2 A_2})(b_1 - b_2)$.

The infinitesimal difference between $\Phi_1(t_1) \circ \Phi_2(t_2)$ and $\Phi_2(t_2) \circ \Phi_1(t_1)$ is

$$[v_1, v_2](\theta) = \frac{\partial}{\partial t_1} \frac{\partial}{\partial t_2}(\Phi_1(t_1) \circ \Phi_2(t_2) - \Phi_2(t_2) \circ \Phi_1(t_1))(\theta) =$$
$$= [A_1, A_2]\theta - A_1 A_2 b_2 + A_2 A_1 b_1 \tag{4}$$

Here $[v_1, v_2]$ stands for *Lie bracket (commutator) of vector fields* and $[A_1, A_2]$ is matrix commutator $A_1 A_2 - A_2 A_1$.

## 3.2 GENERAL CASE

**Theorem 3.1.** *Commutator of the gradient flows for functions $L_1$ and $L_2$ up to second order equals*

$$[\Phi_1(t_1), \Phi_2(t_2)] = t_1 t_2\, R(L_1, L_2) + o(t_1 t_2) \text{ where} \tag{5}$$

$$R(L_1, L_2) = [\nabla L_1, \nabla L_2] = \text{Hess}\, L_2\, \nabla L_1 - \text{Hess}\, L_1\, \nabla L_2 \tag{6}$$

*where* Hess *means Hessian of a function, i.e. matrix of its second derivatives.*

*Proof.* The first part $[\Phi_1(t_1), \Phi_2(t_2)] = t_1 t_2\, [\nabla L_1, \nabla L_2] + o(t_1 t_2)$ is a classic fact from differential geometry (see e.g. (Lee, 2012), also Appendix D.1). The second part $[\nabla L_1, \nabla L_2] = \text{Hess}\, L_2\, \nabla L_1 - \text{Hess}\, L_1\, \nabla L_2$ is the result of direct computation. $\square$

From this, we may answer, when the domain weight schedule $w = (w_k(t))_k$ is locally optimal in class of schedules with fixed total amount of domain data $(\int w_k(t)\mathrm{d}t)_k$. Before doing this, we should define, what kind of optimality do we pursue. Suppose, the overall target is to produce a better model for some loss function $L(\theta)$. As in gradient-based optimization, we then use this function also as a criterion for the training order optimization. Therefore, we call $w = (w_k(t))_k$ *locally optimal* with respect to $L$ if any infinitesimal change in $w$ (in class of schedules with same total domain sizes) would degrade $L$.

**Corollary 3.2.** *(a) Let $L : \Theta \to \mathbb{R}$ be a smooth function. Since the infinitesimal change in domain training order alters $\theta$ in direction $R(L_i, L_j)$, the value $L(\theta)$ would change in direction*

$$P(L_i, L_j; L)(\theta(t)) = \langle R(L_i, L_j)(\theta(t)), \nabla L(\theta(t))\rangle = $$
$$= (\nabla L)^T \left(\text{Hess}\, L_i\, \nabla L_j - \text{Hess}\, L_j\, \nabla L_i\right). \tag{7}$$

*Precisely, let $w = (w_k(t))_k$ and $w^\varepsilon = (w_k^\varepsilon(t))_k$ be weight schedules, such that*

$$w_k^\varepsilon(t) = \begin{cases} w_i(t) + \delta, & \text{if } t \in [t_0, t_0 + \varepsilon) \text{ and } k = i, \\ w_j(t) - \delta, & \text{if } t \in [t_0, t_0 + \varepsilon) \text{ and } k = j, \\ w_i(t) - \delta, & \text{if } t \in [t_0 + \varepsilon, t_0 + 2\varepsilon) \text{ and } k = i, \\ w_j(t) + \delta, & \text{if } t \in [t_0 + \varepsilon, t_0 + 2\varepsilon) \text{ and } k = j, \\ w_k(t), & \text{otherwise,} \end{cases}$$

*then the trajectories $\theta, \theta^\varepsilon$ defined by weight schedules $w, w^\varepsilon$, satisfy*

$$L(\theta^\varepsilon(t_0 + 2\varepsilon)) - L(\theta(t_0 + 2\varepsilon)) = \frac{1}{2}\delta\, \varepsilon^2\, P(L_i - L_j, \Sigma_k w_k(t_0)L_k; L) + o(\delta\, \varepsilon^2). \tag{8}$$

*(b) For any locally optimal training schedule $w = (w_k(t))_k$ with respect to loss function $L$, the corresponding training trajectory $\theta(t)$ satisfies the following condition: for any step $t$ for any $i, j$ if*

$$P(L_i - L_j, \Sigma_k w_k(t)L_k; L)(\theta(t)) \neq 0,$$

*then either $w_i(t) = 0$ or $w_j(t) = 0$.*

*Proof.* Part *(a)* is again a direct computation (follows from the formula for integration of time-dependent ODE, see Appendix D.2).

For part *(b)* suppose the contrary, and without loss of generality, suppose $P(L_i, L_j; L)(\theta(t)) > 0$. Take $w^\varepsilon$ from part *(a)*, with $\delta = \frac{\min(w_i, w_j)}{2} > 0$. This definition implies that $w^\varepsilon$ is right-continuous and non-negative, when $\varepsilon$ is small enough, therefore, it is a valid training schedule. Denote by $\theta^\varepsilon$ the training trajectory of the modified flow. Then, by formula 8, we have $L(\theta^\varepsilon(t + 2\varepsilon)) - L(\theta(t + 2\varepsilon)) = -\frac{1}{2}\varepsilon^2\, \delta\, P(L_i - L_j, \Sigma_k w_k(t)L_k; L)(\theta(t)) + o(\varepsilon^2)$ which is negative for small $\varepsilon$, leading to suboptimality of initial trajectory. $\square$

We note that for the setting with two domains, due to bilinearity and anti-symmetry of the Lie bracket, our criterion says that when $P(L_1, L_2; L) \neq 0$, it is better to train either *mode one:* entirely on the first domain, or *mode two:* entirely on the second; but not to mix them. Moreover, from the (a) part, it also follows that the shift from mode one to training on the second one may occur at time

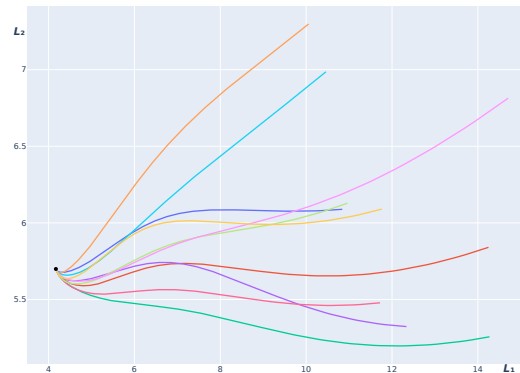

Figure 2: Domain losses dynamics. Here X and Y axes correspond to values of $L_1$ and $L_2$ and curves are trajectories for same constant weight schedule but different initial points. All trajectories converge to a single point which corresponds to the global optimum of $\frac{L_1 + L_2}{2}$, though we see that loss behavior is non-monotonous for some of the trajectories.

$t$ only when $\mathrm{P}(L_1, L_2; L)(\theta(t)) \leq 0$, and conversely, shift from mode two to mode one requires $\mathrm{P}(L_1, L_2; L)(\theta(t)) \geq 0$.

For a given trajectory, if it does not satisfy the optimality criterion from part (b) for some time step $t$, the local change in the order would give a better trajectory.

As it is common, the optimality criterion does not explicitly convert into the algorithm constructing the optimal trajectory. However, it can be applied for analysis an existing training trajectory. Given a series of checkpoints during training, one can check, when the training weight schedule was far from optimal, and make recommendation on the weigh schedule correction. We analyze the predictive power of such recommendations in the following section. Also, potentially, the information from $\mathrm{P}(L_i, L_j; L)$ may be used for online data mixing.

## 4 EXPERIMENTS

### 4.1 QUADRATIC OPTIMIZATION

Let us further analyze the toy example from the previous section. We independently sample two $n \times n$ random matrices $A_{1,2}$ with predefined power law (Xie et al., 2022) spectrum, and random vectors $b_{1,2}$. These datum generates loss functions $L_1$ and $L_2$. The weight schedule $w(t)$ leads to trajectories of the ODE (2). Note that despite the simplicity of this setting, even constant weight schedule may produce non-trivial behavior. For example, the trajectory with constant weights $w_1(t) = w_2(t) = \frac{1}{2}$ may have non-monotonous (see fig. 2) behavior of $L_1(\theta(t))$ and $L_2(\theta(t))$ (of course, $w_1 L_1 + w_2 L_2$ is monotonously decreasing). We want to validate the predictions from Theorem 3.1 on the influence of altering the weight schedule.

We start from constructing the trajectory $\theta(t)$ of gradient descent optimization with constant weight schedule $(0.5, 0.5)$, corresponding to constant loss function

$$L_{basic} = \frac{L_1 + L_2}{2} = \frac{1}{4} \sum_{i=1,2} (\theta - b_i)^T A_i (\theta - b_i)$$

Suppose that we are given parameters $\theta = \theta(t)$ at time $t$ trained with the above weight schedule. Then, we start training weights $(1, 0)$ and $(0, 1)$ (which means training completely on the first and completely on the second domain respectively) for $\Delta t$ steps, obtaining points $\theta_1$ and $\theta_2$ respectively. From these two points, we start training with weights $(0, 1)$ and $(1, 0)$ respectively, obtaining points $\theta_{12}$ and $\theta_{21}$. The total amount of training data used to produce checkpoints $\theta_{12}$, $\theta_{21}$ and $\theta_{base} = \theta(t + 2\Delta t)$ is

Table 1: The predicted vs. observed excess loss, $EL^{12}/P_{L_1,L_2;L}$, for quadratic optimization example.

| $\Delta t$ 
 t | 0.001 | 0.01 | 0.1 |
|---|---|---|---|
| 0.1 | $0.997 \pm 0.001$ | $0.972 \pm 0.010$ | $0.763 \pm 0.053$ |
| 0.3 | $0.997 \pm 0.001$ | $0.973 \pm 0.008$ | $0.766 \pm 0.084$ |
| 1.0 | $0.997 \pm 0.001$ | $0.974 \pm 0.012$ | $0.774 \pm 0.135$ |
| 3.0 | $0.998 \pm 0.002$ | $0.979 \pm 0.022$ | $0.834 \pm 0.223$ |

absolutely the same, but their evaluation would give different results. Theory predicts that

$$L(\theta_{12}) - L(\theta_{base}) \simeq L(\theta_{base}) - L(\theta_{21}) \simeq \frac{1}{8}(\Delta t)^2 \, \mathrm{P}(L_1, L_2; L)(\theta(t)) \simeq$$

$$\simeq \frac{1}{8}(\Delta t)^2 ((\theta - b_1)^T A_1 + (\theta - b_2)^T A_2) \, (A_1 A_2 (\theta - b_2) - A_2 A_1 (\theta - b_1))$$

We take results from training with different starting steps $t$ and intervention lengths $\Delta t$ and compute the values of $L_1$, $L_2$ for $\theta_{12}$, $\theta_{21}$ and $\theta_{base}$. For $i = 1, 2$, we define the excess loss $EL_i^{12} = L_i(\theta_{12}) - L_i(\theta_{base})$, and similarly $EL_i^{21} = L_i(\theta_{21}) - L_i(\theta_{base})$. In Table 1, we compare the predicted and actual values of the excess losses.

## 4.2 BILINGUAL LLM PRE-TRAINING

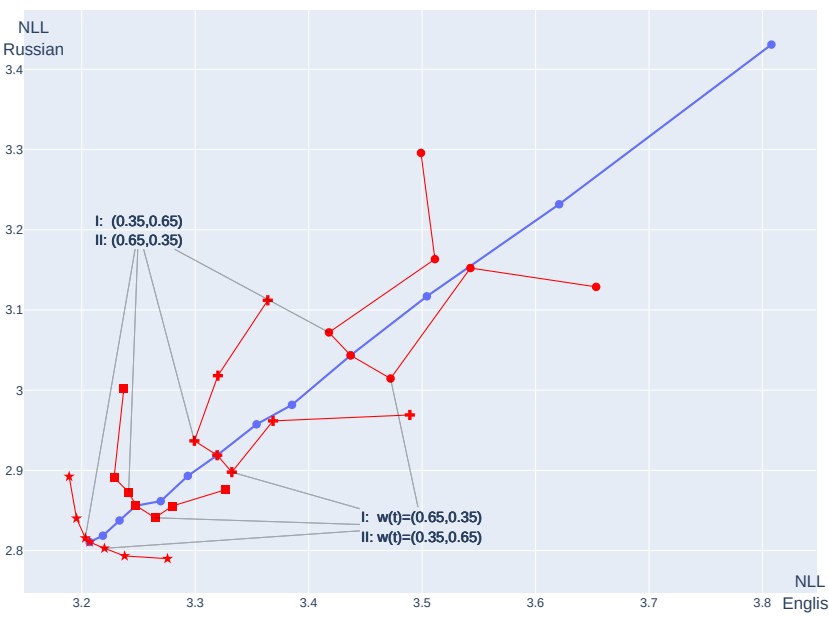

Figure 3: The results of intervening into domain weight schedule for bilingual LLM training. The blue points constitute loss values (negative log likelihoods for English and Russian data) for a training trajectory with constant domain weight schedule $w(t) = (0.5, 0.5)$, from a checkpoint at step 4000 (right, top) to step 28000 (bottom, left). The red points correspond to weight schedules from formula (9). The total amount of English and Russian tokens used for training is constant across each red curve: equal of 5000, 8000, 11000 and 14000 batches for each language for ●, ✚, ■ and ★, respectively; only the degree of intervention $\Delta w$ changes from $0.15$ for annotated points to $0.45$ for the most distant points.

We try to answer, how does the order of training on different domains influences the LLM's performance. To do it, we take a small GPT-2 architecture and pretrain it in two languages. As the

Table 2: Commutation results for LLM. Losses $L_1, L_2$ are NLL on English and Russian domains. Here $EL_i^{12}$ columns mean the excess loss $L_i(\theta_{12}) - L_i(\theta_{base})$ which measures the influence of reordering domains during training. The latter two columns are values $P(L_2, L_1; L_i)$, which (up to rescaling) are the proxy for $EL_i^{21}$ and $-EL_i^{12}$.

| $t$ | $L_1$ | $L_2$ | $EL_1^{12}$ | $EL_2^{12}$ | $EL_1^{21}$ | $EL_2^{21}$ | $P_{2,1;1}$ | $P_{2,1;2}$ |
|---|---|---|---|---|---|---|---|---|
| 2000 | 3.44 | 3.04 | -0.0191 | 0.0286 | 0.0352 | -0.0289 | 0.0195 | -0.0289 |
| 8000 | 3.32 | 2.92 | -0.0201 | 0.0179 | 0.0129 | -0.0211 | -0.0020 | -0.0074 |
| 14000 | 3.25 | 2.86 | -0.0065 | 0.0169 | 0.0168 | -0.0151 | 0.0014 | -0.0048 |
| 20000 | 3.21 | 2.81 | -0.0038 | 0.0049 | 0.0131 | -0.0079 | 0.0049 | -0.0072 |

pre-training data, we take C4 (Raffel et al., 2020) dataset for English, and mC4 (Xue et al., 2020) dataset for Russian. We follow the sampling-based paradigm, i.e. we take weights $(w_1, w_2)$ satisfying $w_1 + w_2 = 1$ and use them as the sampling probabilities for English and Russian datasets to form a mini-batch.

The LLM pre-training is commonly performed using the Adam or AdamW optimizer (see e.g. (Touvron et al., 2023; Bai et al., 2023)) due to its better stability. Despite that our theoretical results are stated and proven for gradient descent (see remark in section 3), we follow the standard pre-training pipeline with Adam. The results in this section show that our results can be applied for training with Adam as well.

Using the same setting as in the previous section, we train a basic model with the weights $(0.5, 0.5)$ for $t_0$ steps and get parameters $\theta_{base} = \theta(t_0)$. Next, starting with the checkpoint $\theta_{base}$, we take $\Delta t$ steps with the weights $(0.5 + \Delta w, 0.5 - \Delta w)$ and $(0.5 - \Delta w, 0.5 + \Delta w)$ in different order, obtaining checkpoints $\theta_{12}$ and $\theta_{21}$. Therefore, checkpoint $\theta_{12}^{t_0, \Delta t, \Delta w}$ is obtained using the following training schedule $w^{12, t_0, \Delta t, \Delta w}(t)$:

$$w_1^{12, t_0, \Delta t, \Delta w}(t) = \begin{cases} 0.5, & \text{if } t \in [0, t_0), \\ 0.5 + \Delta w, & \text{if } t \in [t_0, t_0 + \Delta t), \\ 0.5 - \Delta w, & \text{if } t \in [t_0 + \Delta t, t_0 + 2\Delta t); \end{cases}$$

$$w_2^{12, t_0, \Delta t, \Delta w}(t) = \begin{cases} 0.5, & \text{if } t \in [0, t_0), \\ 0.5 - \Delta w, & \text{if } t \in [t_0, t_0 + \Delta t), \\ 0.5 + \Delta w, & \text{if } t \in [t_0 + \Delta t, t_0 + 2\Delta t). \end{cases} \tag{9}$$

Note that the total number of English tokens and respectively, Russian tokens used for training checkpoints $\theta_{12}, \theta_{21}$ and $\theta(t + 2\Delta t)$ are the same.

Because of the irreducibly noisy nature of stochastic optimization, we use rather large $\Delta t$ which is 4000 training steps with batch size of 512. The starting checkpoints for these interventions are from training step $2000, 8000, 14000, 20000$. We conducted experiments with three options for degree of intervention $\Delta w$: $0.15, 0.3$ and $0.45$. The results are in figure 3. We see that:

- Changes in weight schedule affect the training results even when the total amount of training tokens for both languages is fixed;

- For small initial training step the large interventions have negative impact on training;

- For bigger training steps, the results of training with intervention form well-behaved Pareto fronts;

- Generally, the checkpoint with the modified schedule is better, compared to the baseline, on a language for which we increased the proportion at the end and worse for another language, e.g. $w^{12}$ is better on Russian and worse on English.

Next, we match these results with theoretical predictions. The excess loss formula (7) cannot be computed directly, as the Hessian is computationally intractable. However, the Hessian-vector product can be computed with about the same complexity as the gradient computation (precisely, one needs two times more memory and two times more compute for this operation) (Pearlmutter, 1994; Dagréou et al., 2024). To overcome the noise, we average over 100 batches for domain gradients

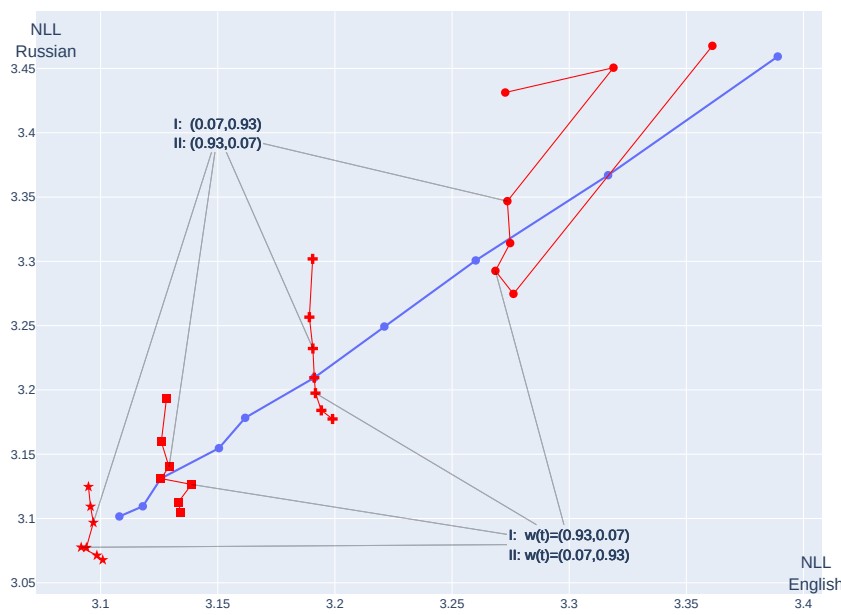

Figure 4: The results of intervening into domain weight schedule for bi-lingual LLM training with language imbalance. The blue points constitute loss values for a training trajectory with constant domain weight schedule $w(t) = (0.9, 0.1)$, from a checkpoint at step 4000 (right, top) to step 25000 (bottom, left). The red points correspond to weight schedules from formula (9). The total amount of English and Russian tokens used for training is constant across each series of points: (9000, 1000), (14400, 1600), (19800, 2200), (24200, 2800) for ●, ✚, ■ and ★, respectively; only the degree of intervention $\Delta w$ changes from 0.03 for annotated points to 0.09 for the most distant points.

computation, and average over 100 batches for further Hessian-vector product computation. To apply results from Section 3, we consider the (discreet time) gradient descent equation $\theta(t + 1) = \theta(t) - \gamma \sum_k w_k \nabla L_k(\theta(t))$, where $\gamma$ is the *learning rate*, as a discretization of ODE 2 with additional factor of $\gamma$ in the right-hand side. Therefore, we adjust $\mathrm{P}(L_1, L_2; L)$ by $\gamma^2$. The results are in table 2. We see that for three of four checkpoints the direction of loss change from the intervention are predicted correctly, though the absolute value is rather noisy. We provide the results of $\mathrm{P}(L_1, L_2; L_i)$ calculation for other checkpoints during training, in table 4 in Appendix.

The discrepancy between the predicted and actual values may come from: (1) the influence of Adam optimizer, (2) the influence of stochasticity and noise in gradient computation, and (3) non-locality: 4000 + 4000 steps for intervention is significant, compared even to the whole length of pre-training (28000 steps for the last checkpoint). Despite, we could not yet check it, we hypothesise that for the checkpoints, where we have both $\mathrm{P}(L_2, L_1; L_1)$ and $\mathrm{P}(L_2, L_1; L_2)$ negative, a more subtle experiment would show that the reordering may have positive effect both of English and Russian performance simultaneously.

*Computational complexity.* We have shown that despite our theoretical analysis is local, the predictions can be still valid on huge time windows (8000 time steps). Therefore, to employ the method in practice (e.g. to apply for online data mixing), one does not need to recalculate $\mathrm{P}(L_i - L_j, \Sigma_k w_k(t)L_k; L)$ too often. The rough estimate gives that if we calculate it once in 1000 steps (these steps are with batch size 512 for this experiment), and take as in this section, averaging over 600 examples for gradient and Hessian-vector product calculation, the computational overhead would be less than half of percent. However, for large amount of domains, it may become significant.

### 4.3 BI-LINGUAL LLM PRE-TRAINING WITH LANGUAGE IMBALANCE

In the previous section we have shown the influence of bi-lingual LLM scores on the domain weight schedule. Also, the theoretical formula for predicting this influence is validated. However, for the

Table 3: Excess losses and predicted excess losses for LLM pre-training with language imbalance.

| $t$ | $L_1$ | $L_2$ | $EL_1^{12}$ | $EL_2^{12}$ | $EL_1^{21}$ | $EL_2^{21}$ | $P_{2,1;1}$ | $P_{2,1;2}$ |
|---|---|---|---|---|---|---|---|---|
| 2000 | 3.26 | 3.70 | -0.0006 | 0.0352 | 0.0037 | -0.0394 | 0.0225 | -0.1454 |
| 8000 | 3.18 | 3.48 | -0.0010 | 0.0216 | 0.0023 | -0.0297 | 0.0053 | -0.0076 |
| 14000 | 3.12 | 3.33 | -0.0012 | 0.0174 | 0.0009 | -0.0233 | 0.0027 | -0.0048 |
| 20000 | 3.09 | 3.22 | -0.0017 | 0.0144 | 0.0007 | -0.0199 | 0.0020 | -0.0317 |

case of equal-sized domains the practical applicability of our method is limited: figure 3 suggests that the order change can cause the shift of the point along the red curves, but it is unclear, whether we could get a point with lower $L = \frac{L_1 + L_2}{2}$.

To address this, we perform a similar experiment but with domain imbalance: the model is trained with 90% of English and 10% of Russian tokens. Exactly the same setting is repeated, checkpoint $\theta_{12}^{t_0, \Delta t, \Delta w}$ is obtained using the following training schedule $w^{12, t_0, \Delta t, \Delta w}(t)$:

$$
w_1^{12, t_0, \Delta t, \Delta w}(t) = \begin{cases} 0.9, & \text{if } t \in [0, t_0), \\ 0.9 + \Delta w, & \text{if } t \in [t_0, t_0 + \Delta t), \\ 0.9 - \Delta w, & \text{if } t \in [t_0 + \Delta t, t_0 + 2\Delta t); \end{cases}
$$
$$
w_2^{12, t_0, \Delta t, \Delta w}(t) = \begin{cases} 0.1, & \text{if } t \in [0, t_0), \\ 0.1 - \Delta w, & \text{if } t \in [t_0, t_0 + \Delta t), \\ 0.1 + \Delta w, & \text{if } t \in [t_0 + \Delta t, t_0 + 2\Delta t). \end{cases}
$$

$$(10)$$

The options for the degree of intervention $\Delta w$ are $0.03, 0.06$ and $0.09$. We show the results in figure 4 and match them with predicted in table 3.

It can be seen that the points in each group form almost vertical lines (with presence of outliers). It means that by changing the training order we could get a significant increase in low-resource language performance having negligible decrease in the high-resource language performance. Therefore, our theory and this experiment justify that in the situation of language imbalance, it is better to increase the proportion of low-resource language only at the end of the pre-training.

## 5 RELATED WORK

The multi-domain learning addresses the problem of designing a model such that it can properly handle examples coming from different data sources with varied characteristics, thus, improving its generalization or applicability. To learn simultaneously on several domains, we need to solve a multi-objective problem (Maurer et al., 2016; Désidéri, 2012) or average the losses across domains with some weights $w_k$ (Kokkinos, 2017). The classic approach to choosing the weights is scalarization, where weights are set to some constants. However, the choice of these constants is generally a hard optimization problem (Royer et al., 2024). Another option is adaptive methods that dynamically adapt the weights, i.e. use some weight schedule during training. These methods are divided into loss-based and gradient-based. Loss-based methods compute $w_k(t)$ to make the training uniform among different losses, see e.g. (Liu et al., 2021). Gradient-based methods use gradient information for each domain to re-compute weights needing several backward operations for each step, see e.g. (Javaloy & Valera, 2021; Lin et al., 2019; Chen et al., 2018; 2020).

In our work, we rely on the Lie bracket computation of loss vector fields to conduct our analysis. It was previously used by (Dherin, 2023) for backward error analysis in a continual learning scenario. The work (Marcotte et al., 2024) in some sense complements our work by trying to find, what is invariant (rather than changed) under the altering in domain weights. To do it, they construct a Lie algebra generated by domain gradient vector fields and find the conservation laws. The computation of the Lie bracket in our case leads to the need for Hesian-vector product computation. We note that Hessian computations occur in various fields of Machine Learning, including optimization (Liu & Nocedal, 1989; Nocedal & Wright, 1999), loss landscape analysis (Cooper, 2018), uncertainty

estimation (Daxberger et al., 2021), and bi-level optimization (Chen et al., 2022; Franceschi et al., 2018).

The influence of data from different domains on a model's performance is also an active research topic. The common approach here is to utilize influence functions (Koh & Liang, 2017), or some of their approximations, like TracIn (Pruthi et al., 2020; Schioppa et al., 2022). Note that despite the visual similarity of the influence formula (Koh & Liang, 2017) and our formula 8, the difference is that our formula has Hessian rather than inverse Hessian.

## 6 CONCLUSION

With the rise of Large Language Models (LLM), it becomes clear that the data is crucial to the LLMs' success. Much research is devoted to analyzing best practices for dataset collection, namely cleaning, filtering, domain and language proportion, etc. Much less attention is paid to the training order, while practical evidence shows that the dataset order matters. We believe that one reason is that the researchers usually use the simplified theoretical understanding of the training process, supposing that the data is sampled from the overall data distribution throughout training. Meanwhile, in the case of the highly long training process on the diverse multi-domain multi-lingual dataset, distribution stability is unlikely to be achieved. Therefore, a novel theoretical framework closer to the realistic setting is needed. In this work, we step in this direction, proposing a theoretically motivated method to analyze and predict the influence of the training order in a multi-domain setup. We demonstrate the soundness of our ideas in both toy examples and realistic experiments. Our results on multi-lingual LLM pre-training illustrate the importance of the raised problem and the ability of our theoretical framework to explain the observed variation in the training results depending on the dataset order only.

### 6.1 LIMITATIONS

Firstly, the method does not give an explicit algorithm to produce an optimal weight schedule. It just answers, to which direction should we shift an existing sub-optimal one to improve it.

Secondly, our method currently does not handle the difference between optimizers, see remark in section 3. However, it is known that the choice of the optimizer can influence the training dynamics and lead to regions with different loss landscape (Jiang et al., 2024). Modifications of optimizers for multi-domain and multi-task learning (Yang et al., 2023) also would change the commutator formulae.

More importantly, we currently do not properly handle the stochasticity of the training process. The training process is better described as a flow of stochastic differential equation $\mathrm{d}\theta = -\nabla L(\theta)\,\mathrm{d}t + \Omega^{\frac{1}{2}}(\theta)\,\mathrm{d}W$, where $W$ is the Wiener process. Here $\Omega(\theta)$ measures the noise intensity of the stochastic gradient and is often crucial for learning. As we have shown, the infinitesimal difference in the order of different domains would result in bias in the drift term (Theorem 3.1), but also it would change noise intensity in the diffusion term. The formal derivation of this dependency (using Ito or Stratonovitch paradigms, and Chen-Strichartz formula, see e.g. (Baudoin, 2004)), as well as its empirical validation, is our further direction.

## 7 REPRODUCIBILITY STATEMENT

For our example with quadratic optimization, the experiment details are in Appendix B. Also, we provide all the necessary code to reproduce.

Details of LLM pre-training experiment are in Appendix C. We do not provide the complete pre-training codebase (based on Megatron-Deepspeed repository) because of it's irrelevance, but provide the scripts for the training interventions, for reference. The code for checkpoint analysis (including Hessian-vector product formulae) is provided.

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

Table 4: $P_{12;i}$ mean the predicted infinitesimal excess loss $P(L_1, L_2; L_i)$.

| $t$ | $P_{12;1}$ | $P_{12;2}$ |
|---|---|---|
| 2000 | 0.0195 | -0.0289 |
| 4000 | 0.0056 | -0.0344 |
| 6000 | -0.0001 | -0.0127 |
| 8000 | -0.0020 | -0.0074 |
| 10000 | 0.0010 | -0.0038 |
| 12000 | 0.0009 | -0.0050 |
| 14000 | 0.0014 | -0.0048 |
| 16000 | 0.0040 | -0.0039 |
| 18000 | -0.0054 | -0.0067 |
| 20000 | 0.0049 | -0.0072 |
| 22000 | -0.0043 | -0.0149 |
| 24000 | -0.0004 | -0.0141 |

(a) balanced domains

| $t$ | $P_{12;1}$ | $P_{12;2}$ |
|---|---|---|
| 2000 | 0.0225 | -0.1454 |
| 4000 | 0.0059 | -0.0600 |
| 6000 | -0.0233 | -0.0216 |
| 8000 | 0.0053 | -0.0076 |
| 10000 | 0.0043 | -0.0167 |
| 12000 | -0.0029 | -0.0080 |
| 14000 | 0.0027 | -0.0048 |
| 16000 | 0.0106 | -0.0184 |
| 18000 | 0.0004 | -0.0221 |
| 20000 | 0.0020 | -0.0317 |
| 22000 | -0.0040 | -0.0226 |
| 24000 | 0.0060 | -0.0396 |

(b) domain imbalance

L Xue, N Constant, A Roberts, M Kale, R Al-Rfou, A Siddhant, A Barua, and C Raffel. mt5: A massively multilingual pre-trained text-to-text transformer. *arXiv preprint arXiv:2010.11934*, 2020.

Enneng Yang, Junwei Pan, Ximei Wang, Haibin Yu, Li Shen, Xihua Chen, Lei Xiao, Jie Jiang, and Guibing Guo. Adatask: A task-aware adaptive learning rate approach to multi-task learning. In *Proceedings of the AAAI conference on artificial intelligence*, volume 37, pp. 10745–10753, 2023.

Tianhe Yu, Saurabh Kumar, Abhishek Gupta, Sergey Levine, Karol Hausman, and Chelsea Finn. Gradient surgery for multi-task learning. In H. Larochelle, M. Ranzato, R. Hadsell, M.F. Balcan, and H. Lin (eds.), *Advances in Neural Information Processing Systems*, volume 33, pp. 5824–5836. Curran Associates, Inc., 2020. URL https://proceedings.neurips.cc/paper_files/paper/2020/file/3fe78a8acf5fda99de95303940a2420c-Paper.pdf.

Zhan Zhuang, Yu Zhang, and Ying Wei. Gradual domain adaptation via gradient flow. In *The Twelfth International Conference on Learning Representations*, 2024. URL https://openreview.net/forum?id=iTTZFKrlGV.

# A    ADDITIONAL RESULTS FOR BI-LINGUAL LLM

Predicted excess loss for other checkpoints of bi-lingual LLM pre-training (experiments from sections 4.2,4.3) are in table 4.

# B    TOY EXAMPLE DETAILS

To sample matrices $A_1, A_2$, we first determine their dimension 100, and put eigenvalues $\Lambda = (\lambda_j)_{j=0}^{99}$ according to power law $\lambda_j = 0.7^j$, and then conjugate them with a random orthogonal matrices $C_1, C_2$: $A_i = C_i^T \operatorname{Diag}(\Lambda) C$. Vectors $b_1, b_2$ are sampled from standard Gaussian distribution with unit standard deviation in each direction.

For table 1, we sample starting point $\theta$ from Gaussian distribution, and analytically compute the flow of ODE 2 for $L = \frac{L_1 + L_2}{2}$ by time $t$ and $t + 2\Delta t$. Then we compute flow for similar ODE for loss functions $L_1, L_2$ and apply them in order to get $\theta_{12}(t)$. Excess loss will be the difference $L(\theta_{12}(t)) - L(\theta(t + 2\Delta t))$. We compare it with prediction $P(L_1, L_2; L)$. In the table we present the median and $90 - 10$ percentile range.

## C LLM PRE-TRAINING DETAILS

Our setup follows classical GPT-2 model pre-training, using Megatron-Deepspeed repository[1]. C4 and mC4 datasets are publicly available. We used mBERT tokenizer. Training was performed with global batch size 512, and sequence length 1024. We used Adam optimizer with $\beta_1 = 0.9$ and $\beta_2 = 0.999$. The learning rate was set to $1.5 \times 10^{-4}$.

The Hessian-vector products were calculated using reverse-on-reverse method from (Dagréou et al., 2024). For the final values of gradients and Hessian-vector products, we averaged over 600 random samples for each domain.

## D FORMULAE DERIVATION

### D.1 PROOF OF THEOREM

$$\Phi_1(t_1)x = x + t_1 v_1(x) + \frac{t_1^2}{2}\dot{v}_1(x)v_1(x) + o(t_1^2)$$

$$\Phi_2(t_2) \circ \Phi_1(t_1)x = \Phi_2(t_2)(x + t_1 v_1(x) + \frac{t_1^2}{2}\dot{v}_1(x)v_1(x) + o(t_1^2))$$

$$= x + t_1 v_1(x) + \frac{t_1^2}{2}\dot{v}_1(x)v_1(x) + o(t_1^2) +$$

$$+ t_2 v_2(x + t_1 v_1(x) + \frac{t_1^2}{2}\dot{v}_1(x)v(x) + o(t_1^2)) +$$

$$+ \frac{t_2^2}{2}\dot{v}_2(x + t_1 v_1(x) + \frac{t_1^2}{2}\dot{v}_1(x)v(x) + o(t_1^2)) \cdot$$

$$\cdot v_2(x + t_1 v_1(x) + \frac{t_1^2}{2}\dot{v}_1(x)v(x) + o(t_1^2)) + o(t_2^2) =$$

$$= x + t_1 v_1(x) + \frac{t_1^2}{2}\dot{v}_1(x)v_1(x) + o(t_1^2) +$$

$$+ t_2 v_2(x) + t_1 t_2 \dot{v}_2(x)v_1(x) + o(t_1 t_2) +$$

$$+ \frac{t_2^2}{2}\dot{v}_2(x)v_2(x) + o(t_2^2)$$

$$\Phi_1(t_1) \circ \Phi_2(t_2)x - \Phi_2(t_2) \circ \Phi_1(t_1)x = t_1 t_2(\dot{v}_1(x)v_2(x) - \dot{v}_2(x)v_1(x)) =$$
$$= t_1 t_2(\text{Hess } L_1 \nabla L_2 - \text{Hess } L_2 \nabla L_1) + o(t_1 t_2)$$

### D.2 INTEGRATION FORMULA FOR COROLLARY 3.2(A)

$$\theta(t_0 + \tau) = \theta(t_0) + \sum_k \left( \int_0^\tau w_k(t_0 + \tau_1)d\tau_1 \right) \nabla L_k(\theta_0) +$$

$$+ \sum_{k_1, k_2} \left( \int_0^\tau \int_0^{\tau_1} w_{k_1}(t_0 + \tau_1)w_{k_2}(t_0 + \tau_2)d\tau_1 d\tau_2 \right) \text{Hess } L_{k_1} \nabla L_{k_2} + o(\tau^2)$$

---

[1]https://github.com/microsoft/Megatron-DeepSpeed

