# OpenReview forum: "Commute Your Domains: Trajectory Optimality Criterion for Multi-Domain Learning"
_ICLR.cc/2025/Conference — Submitted to ICLR 2025_

### Official Review · Reviewer_vKQ3 · 2024-10-30

**Soundness:** 2
**Presentation:** 3
**Contribution:** 2
**Rating:** 3
**Confidence:** 4

**Summary:**

The order of adaptation domains used for training is important. The paper considers a scenario where the amount of data is fixed and the examples from different domains can be adjusted. They study the training order using a theoretical framework of a toy example and bilingual LLM pre-training. Overall, there are some limitations to the current version, I would like to raise the score if these limitations can be solved in the feedback.

**Strengths:**

1. The paper investigates an interesting idea that is important for multi-domain learning.
2. The paper proposes a theoretical framework to analyze the influence of the training order.
3. The paper is written clearly and well-organized.

**Weaknesses:**

Just as the authors stated in the limitation, there are still some weaknesses:

1. There are no concrete algorithms to solve the problems proposed in the paper. There are only some insights from the paper.

2. The experimental results are not strong enough. I am worried the conclusions can be generalized to LLM models, especially for other multi-domain scenarios.

**Questions:**

1. The introduction is a little short. This makes the motivations of the paper not strong enough. refer to https://arxiv.org/pdf/2104.08786

2. Although the order is important for multi-domain setup, there is still a "catastrophic forgetting" issue. In this case, I am not sure if it is necessary to target "order" analysis in the multi-domain. Could you explain a little bit if the theoretical analysis in the paper is useful for catastrophic forgetting issues?

3. There are only bilingual experimental results in the paper, could you show some results based on 5 more languages or multi-tasks results? refer to the paper:https://arxiv.org/abs/2405.11157.

4. GPT2 model is a "smaller" model, I would like to see some experimental results based on LLama-7b or 13b.

---

> ### Author Response · Authors · 2024-11-23
>
> Dear Reviewer vKQ3,
>
> Thank you for your time and effort in this thorough review!
>
> Concerning the practical use, here we propose potential applications. Based on the theory, we may examine the optimality of the training trajectory, thus, making an analysis of different mixing algorithms and provide recommendation on adjusting the mixing weights near a given checkpoint. Moreover, we can compute the P-value of the potential weight intervention during the training and locally change the mixing proportions. Based on the P-value, we may decide for which domains it's better to increase their proportion at the end and decrease at the start.
>
> Validation on the other multi-domain scenarios is an interesting direction for the further research. In our paper, we were concerned with presenting new method for the LLMs training analysis and show some examples on it. We mainly target the new theoretical framework to analyze the ordering of domains during the training. We took  a two-languages as minimal example. We believe that results would generalize to other languages and amount of domains.
>
> Some additional experimental results are obtained for the same bi-lingual LLM setup but with domain (i.e. language) imbalance (90% of first language and 10% of second). In this case results show that the effect of order on the loss in high-resource language is much less than the effect on the loss in low-resource language. Therefore, we can benefit the low-resource one without significant draw-down in the high-resource one. The results will be added to the paper. We were unable to perform similar experiments with 7b and larger models, since the pre-training is computationally expensive, but we believe that the predictive power of the proposed theory persists.
>
> The introduction section indeed does not cover some of relevant works. We added some of them in the next version. In particular, we provide references to some studies ignoring the training order and dealing only with the amount of data. The paper on the order of the in-context examples seems quite interesting. Despite that the authors of this paper do not deal with any kind of training, the in-context leaning is still learning, and also benefits from the correct order (of course, by means of different mechanisms).
>
> Thanks again for your insightful feedback. We will improve the text accordingly.

---

> > ### Comment · Reviewer_vKQ3 · 2024-11-26
> > **reviews**
> >
> > Thanks for your feedback. I agree there are some insightful findings in the paper. However, there needs more experimental analysis to prove it is necessary to focus on the "order" rather than "routing". Also, gpt2 is a very "small" model, if it is a problem to use 7b or larger models, it is fine to evaluate "smaller" models based on different architectures.
> >
> > I will keep my score since there is no additional experimental study as I expected.

---

### Official Review · Reviewer_kMLr · 2024-10-31

**Soundness:** 3
**Presentation:** 3
**Contribution:** 3
**Rating:** 6
**Confidence:** 2

**Summary:**

The paper investigates the effects of training order on model performance in multi-domain learning contexts, where a model learns from diverse data sources. Recognizing that the sequence of domain exposure can significantly impact outcomes, the authors propose a theoretical framework using the Lie bracket of gradient vector fields. This framework identifies areas in parameter space where modifying the training order may improve target loss. The authors demonstrate the theoretical framework's predictions with both a controlled "toy" example and a bilingual large language model (LLM) pre-training task, providing insights into optimizing training order for enhanced performance across domains.

**Strengths:**

1. The paper introduces an original theoretical approach to training order optimization in multi-domain learning, leveraging Lie bracket analysis of gradient vector fields.

2. Methodologically robust, the paper validates its theoretical insights through both synthetic and realistic experiments, particularly a bilingual LLM pre-training task.

3. The work is significant for practical and theoretical advancements.

**Weaknesses:**

1. The paper assumes gradient and Hessian computations that may not account for stochasticity and optimizers like Adam, which is commonly used in deep learning and could affect convergence behavior. This might lead to inaccuracies in predicting training outcomes.

2. While the theoretical framework using Lie brackets provides insight into training order in multi-domain learning, it lacks direct, actionable guidance for practitioners. The method suggests a direction for optimizing the training sequence but doesn’t provide a concrete algorithm for determining an optimal sequence.

**Questions:**

1. Do you have suggestions for efficient approximation methods for Hessian calculations, or could you discuss any empirical limits encountered in terms of model size and domain count?

2. Could you address the potential discrepancy introduced by stochastic gradients and adaptive optimizers in practical applications?

---

> ### Author Response · Authors · 2024-11-23
>
> Dear Reviewer kMLr,
>
> Thank you for your time and effort in this thorough review!
>
> We train our bilingual models with Adam optimizer and average domain gradients over several batches to remove the noise due to the stochasticity. According to our empirical results, the theoretical predictions give reasonable estimations as generally we are only interested in the sign of the examined quantity (loss became better or not).
> Introducing the stochasticity requires averaging over multiple examples during the gradients and Hessian-vector products computation. The remaining discrepancy coming from the commutation of the stochastic differential operators (Limitations section of our paper) is hard to estimate, it is the topic for the following research. Initial guess is that theoretically, this additional term would not be significant. To make actual estimates, more experiments with simplified models of data should be performed.
>
> As you mention, the method, as it is stated, suggests a direction for altering the existing training sequence. Potentially, we hope that our method can be used online during the training by making look-ahead predictions for different training schedules. Currently we have no experimental validation of this approach.
>
> Concerning the empirical limits and practical implementation of Hessian calculation, as we mention in the end on section 4, for a given batch of data, the Hessian-vector product (HVP) can be caclulated with twice as much compute and twice as much memory compared to gradient computation. Our formula for mutual influence of two domains requires two HVPs. Therefore, no limitations on model size are imposed. To reduce noise we used averaging over 100 batches. For multiple domains -- calculation of all pairwise influences would require n(n-1)/2 such computations. However, for determining the best direction to shift an existing weight schedule, we would not need all pairwise mutual influences, only n computations would be needed.
>
> Thanks again for your thorough and constructive comments.

---

> > ### Comment · Reviewer_kMLr · 2024-12-01
> >
> > Thank you for your detailed response and clarifications. I appreciate the additional context regarding the stochastic elements and the constraints for Hessian-vector product calculations. While I understand the theoretical framework and its limitations, I still find that the practical guidance for determining optimal training sequences could be more explicit, as this would significantly enhance the applicability of your approach. I will maintain my score and look forward to seeing further developments in this area.

---

### Official Review · Reviewer_61Cp · 2024-10-31

**Soundness:** 3
**Presentation:** 3
**Contribution:** 1
**Rating:** 3
**Confidence:** 3

**Summary:**

The paper studies the problem of training orders in multi-domain learning. The authors introduce a theoretical framework based on the concept of Lie brackets of gradient vector fields to predict how changes in training order can influence model performance across domains. The theoretical insight is validated empirically through both a toy example and bilingual large language model (LLM) pre-training.

**Strengths:**

1. The theoretical framework is well formulated, and the illustrations are intuitive.

2. The writing is clear and structured.

**Weaknesses:**

1. The authors do not provide a clear explanation about how the studied problem is different from the rich literature of multi-task learning (MTL). In MTL, there exists many methods to balance the training of data from different mixtures, and many of them can be provably applied to reach the desired optimum based on loss combinations. It is unclear how results in the paper are different from those.

2. The utility of the theoretical results is limited. The theoretical part essentially provides a way to predict the performance given weight schedule. However, this does not provide a very accurate prediction due to the stochastic nature of optimization, and the computational cost is non-negligible.

**Questions:**

I wonder if the authors could provide some comments about the connection with the literature of gradual domain adaptation (GDA) [1,2,3]? The idea is quite relevant, as GDA studies the problem of gradual distribution shift from the source to target domains. There exists algorithms [2,3] that interpolate between source and target domains to construct a path along which the model iteratively improves. This is similar to the gradually changing mixture weights mentioned in the paper.

[1] Understanding Self-Training for Gradual Domain Adaptation. Kumar et al. 2020.
[2] Gradual Domain Adaptation: Theory and Algorithms. He et al. 2023.
[3] Gradual Domain Adaptation via Gradient Flow. Zhuang et al. 2024.

---

> ### Author Response · Authors · 2024-11-23
>
> Dear Reviewer 61Cp,
>
> Thank you for your time and effort in this thorough review!
>
> You note that the paper lacks the description of the difference between our approach and classical works on multi-task learning. Actually, the main difference is the addressed problem itself: we study the influence of the training order given the fixed amount of data in every domain. To the best of our knowledge, we give the first theoretical study of this influence. Since it is indeed not clear from the text, we will add the discussion of relevant topics in multi-domain and multi-task learning to the introduction section.
>
> Concerning the noise level (stochasticity) and cost of the additional computations, indeed Hessian-vector products introduce additional complexity. As we state in the paper, this calculation requires two times more memory and four times more compute than ordinary gradient computation. However, we do it infrequently as in practice the changes to the mixing weights would be applied for >1K steps ahead. In this case, 600 (as in our experiment, 100 batches of size 6) Hessian-vector products for 512K (1K steps of batch size 512) gradient computation would give practically negligible computation overhead. We would add these calculation to the paper.
>
> We note that the presented theory can be used to examine existing methods for data mixing on their optimality. We can say which parts of the trajectory aren't optimal. Based on the predictions, we can make changes to the mixing preserving the amount of the data for each domain to improve the model's performance. For simplicity, we've used bi-lingual LLM task and validated on the same distribution, but, it is common to have validation domain that differs from the training. For example, LLMs are trained on some mixture of data and then evaluated on the downstream tasks. In this case, we can re-order data to improve the downstream performance.
>
> Your questions on gradual domain adaptation are insightful. Indeed, since our method is general enough to be applied to any setup with two or more domains, GDA, among others, could be a good application. Optimizing the data mixing from intermediate pseudo-domains using our method is an interesting research topic.
>
> Thanks again for highlighting your concerns about the parameters. If you consider the explanations helpful, we shall improve the manuscript accordingly.

---

> > ### Comment · Reviewer_61Cp · 2024-11-26
> >
> > Thank you for your response. For the first weakness, the authors can consider updating their manuscript with a discussion of relevant topics. Otherwise, it is difficult to assess whether the first point of weakness can be addressed. In addition, I still do not see a convincing practical application of the theoretical results, especially given its non-negligible and non-scalable implementation. Thus, I will maintain my score.

---

### Official Review · Reviewer_Xgd4 · 2024-11-04

**Soundness:** 2
**Presentation:** 2
**Contribution:** 2
**Rating:** 5
**Confidence:** 2

**Summary:**

This paper studied how the order of training on different data domains affects model performance in multi-domain setting. The authors develop a theoretical framework based on Lie bracket analysis of gradient vector fields to predict and understand the effects of changing domain training order. They introduce a trajectory optimality criterion that helps determine when to switch between domains during training. The framework is validated through experiments on both a toy quadratic optimization problem and a bilingual language model pre-training task.

**Strengths:**

1. The paper provides a theoretical foundation for analyzing domain ordering effects in multi-domain learning, an important but under-studied problem.
2. The use of Lie bracket analysis is interesting and novel. Through the analyze of the commutable property of the gradient flow, the authors show the effect of different ordering.
3. The theoretical framework successfully predicts directional changes in loss values when domain ordering is modified, as demonstrated in both synthetic and real-world experiments.
4. The authors also provide clear geometric intuitions for their results through visualizations.

**Weaknesses:**

1. While the theoretical framework can predict the effects of changing domain order, it doesn't provide an explicit algorithm for finding optimal domain schedules.

2. the current theory doesn't fully account for the effects of different optimizers (like Adam) or the stochastic nature of training, which are crucial in deep learning. The experiments, while supportive of the theory, show some discrepancies between predicted and actual values, particularly in the LLM pre-training case.

3. The paper's analysis is limited to two-domain scenarios, and it's not clear how well the approach scales to settings with many domains.

4. The practical applicability of the method may be limited by the computational cost of computing since the involving of the Hessian-vector products.

5. Presentation issue. Please use the correct citation command, for example \citep.

6. What are the key practical message that machine learning practitioner can use from the work?

7. A general question that the author could consider.  How does curriculum learning, where the ordering of training examples is crucial, related to the work?

**Questions:**

See the weakness section for details.

---

> ### Author Response · Authors · 2024-11-23
>
> Dear Reviewer Xgd4,
>
> Thank you for your time and effort in this thorough review!
>
> Presentation (citation) issues are fixed.
>
> Just as you stated, the theory is formulated for the gradient descent, despite that in practical deep learning other optimizers are used. We discuss the discrepancy in Remark in section 3 and also in the limitation subsection. However our experiment with training bi-lingual models was conducted using Adam optimizer. Though it was not proven theoretically, the update direction in Adam is derived from gradients in the points of training trajectory, and therefore we believe that gradient descent could in our case be a valid approximation.
>
> Concerning the cost of the additional computations, as you mentioned, Hessian-vector products introduce additional complexity. As we state in the paper, this calculation required two times more memory and four times more compute than ordinary gradient computation. However, we do it infrequently as in practice the changes to the mixing weights would be applied for >1K steps ahead. In this case, 600 (as in our experiment, 100 batches of size 6) Hessian-vector products for 512K (1K steps of batch size 512) gradient computation would give practically negligible computation overhead. We will add these calculation to the paper.
>
> Concerning the question on two-domain scenarios, we should say that the method and its theoretical explanation works well for any number of domains. We chosen two-domain setups for toy example and bilingual LLM for simplicity and clarity. The Corollary 3.2 is stated for any number of domains and states that for an optimal trajectory at each step any two domain gradients either commute or the weights of some domains are zero. We will add some words to our theory section, since it was indeed not clear from the text.
>
> We note that the presented theory can be used to examine existing methods for data mixing on their optimality. We can say which parts of the trajectory aren't optimal. Based on the predictions, we can make changes to the mixing preserving the amount of the data for each domain to improve the model's performance. For simplicity, we've used bi-lingual LLM task and validated on the same distribution, but, it is common to have validation domain that differs from the training. For example, LLMs are trained on some mixture of data and then evaluated on the downstream tasks. In this case, we can re-order data to improve the downstream performance.
>
> The curriculum learning is an interesting direction of the further research. Currently, we fix the amount of data for each domain and make only local changes to the schedule. But, in curriculum learning it's more interesting to make global changes and stop learning on some of the domains if they are easy to learn. We will add the topic of curriculum learning to the list of relevant topics in the introductory section.
>
> Thanks again for the valuable ideas and comments.

---

> > ### Comment · Reviewer_Xgd4 · 2024-11-25
> >
> > I remain my score unchanged. See reasons below.
> >
> > About Q4: '''this calculation required two times more memory and four times more compute than ordinary gradient computation.'''
> >
> > This extra memory and computational burden is not negligible. I am not convinced that model developers are willing to take this extra burden especially for models with large parameters size.
> >
> > About the response '''...Based on the predictions, we can make changes to the mixing preserving the amount of the data for each domain to improve the model's performance....'
> >
> > Mixing ratio, mixing frequency, mixing sources are all open questions being explored. The response is too general to be convincing. I am not sure what are the new insights/perspectives that the work can bring for empirical researchers.

---

### Official Review · Reviewer_zRLt · 2024-11-04

**Soundness:** 3
**Presentation:** 3
**Contribution:** 3
**Rating:** 5
**Confidence:** 2

**Summary:**

This paper studies the effect of domain orders in the multi-domain learning problems. With the lens of vector field, it shows that the order of domain influences training dynamics. Furthermore, it proposes scheduling for weights to sample batch of each domain, which can benefit the target loss. Finally, it validates its theory with numerical experiments.

**Strengths:**

Disclaimer: I do not have proper knowledge to evaluate its theoretical analysis. It is hard to judge the significance of the theories the paper has provided.

- Provide theoretical analysis about effect of domain order.

- Propose scheduling for weight to sample domain batches grounded on the theory.

- Validate the theoretical analysis with the numerical experiments.

**Weaknesses:**

- Hard to tell actual benefits of the proposed weight scheduling. Based on Figure 3, the constant domain weight schedule seems to work well. Better to elaborate the practical advantage of the proposed method.

- I think there are many relevant works. The final goal is to learn to minimize the total domain loss without interfering other domains, which is the goal of multi-task learning. It would be better to compare the proposed method against some well-known multi-task learning methods (such as [1,2,3,4]) and show its benefit compared to them.


[1] Navon, Aviv, et al. "Multi-Task Learning as a Bargaining Game." International Conference on Machine Learning. PMLR, 2022.

[2] Lee, Seanie, et al. "Sequential Reptile: Inter-Task Gradient Alignment for Multilingual Learning." International Conference on Learning Representations. 2022.

[3] Yu, Tianhe, et al. "Gradient surgery for multi-task learning." Advances in Neural Information Processing Systems 33 (2020): 5824-5836.

[4] Wang, Zirui, et al. "Gradient Vaccine: Investigating and Improving Multi-task Optimization in Massively Multilingual Models." International Conference on Learning Representations. 2021.

**Questions:**

- What is the benefit of using the proposed weight scheduling method? Does it converge faster or converge to better optima?

---

> ### Author Response · Authors · 2024-11-23
>
> Dear Reviewer zRLt,
>
> Thank you for your time and effort in this thorough review!
>
> First of all, with the presented approach, we show that the data ordering can't be neglected during the training and making small interference into the pre-specified mixing weights, we can change the model's performance forming Pareto front. Also, we showed that if we interfere early in the training, it can severely degrade the performance. There are some works like [1] that fix the amount of examples per-domain and use some kind of a sampling method to form a schedule, but they do not account for the ordering of examples. With our approach, we can analyse such methods on their optimality.
>
> Concerning the potential benefits of changing the weight schedule: for the mentioned setup with bi-lingual model training with equal data budget in each language, we agree that constant schedule works well if the target is also equal-weight scalarization of two language losses. However, for non-equal target loss, other points of our Pareto front would be better. More importantly, we will add the results of the similar experiment with bi-lingual LLM pre-training, when we have significant data imbalance 90% of one language and 10% of another. In this case results show that the effect of order on the loss in high-resource language is much less than the effect on the loss in low-resource language. Therefore, we can benefit the low-resource one without significant drawdown in the high-resource one. This is an example of benefit from the proposed approach.
>
> Another practical application of the study is the training regime were the order of the domains is changed based on the predictions from the theory. As experiments with bi-lingual model suggest we can do that infrequently and the estimation is still quite reasonable due to the averaging effects.
>
> Thank you for the provided list of papers relevant to multi-task learning. We will cite them in the work since they would help to create a general picture of the field.  However, in our work we have a specific setting of analyzing the influence of data mixing given the fixed total data budget in all domains. The gradient manipulation (e.g. PCGrad) and even rescaling (e.g. "Multi-Task Learning as a Bargaining Game") are not strictly falling to our setup.
>
> [1] UniMax: Fairer and more Effective Language Sampling for Large-Scale Multilingual Pretraining

---

> ### Comment · Reviewer_zRLt · 2024-11-25
>
> Thank you for the answers. I am not still convinced of the practical applications of the proposed method. Without any numerical experiments to show its use-case, it is hard for me as an  empirical researcher to validate the practical applications.

---

### Author Response · Authors · 2024-11-29

Dear Reviewers,

Please check the updated version.

Some of the mentioned issues are eliminated.

- In particular, the related works section in the introduction is improved.

- Also, the results of second experiment with bi-lingual LLM pre-training is added. It shows the practical gain from changing the weight schedule in domain imbalance situation.

- The statement of corollary is changed due to mathematical mistake. The previous formulation was correct for 2 domains, but not generally.

Thanks for your work on reviewing our paper!

---

### Meta-Review · Area_Chair_Vrp7 · 2024-12-22

**Metareview:**

**Summary:** This paper investigates the effect of training order in multi-domain learning using Lie brackets of gradient vector fields as a theoretical framework. It identifies how modifying domain order influences training dynamics and suggests potential benefits of adjusting domain mixing weights. The paper presents theoretical insights validated on a toy example and bilingual LLM pre-training.

**Decision:** While this paper proposes a novel theoretical analysis of domain ordering in multi-domain learning, it falls short in several critical areas. Notably, the framework lacks actionable algorithms or practical guidance, limiting its utility for practitioners. In addition, reviewers also raise concerns about empirical results and analysis to validate the scalability and generalization of the results to larger models or more diverse scenarios. Additionally, the computational overhead introduced by Hessian-vector product calculations and the challenges posed by stochastic optimization further diminish its feasibility for real-world applications. The paper also does not sufficiently distinguish itself from existing works in multi-task learning and related fields like gradual domain adaptation, leaving its contribution to the literature unclear. These limitations collectively outweigh the paper's theoretical contributions, leading to the decision to reject. During the reviewer-AC discussion period, no objections were raised to this decision.

**Additional Comments On Reviewer Discussion:**

During the rebuttal phase, reviewers maintained their primary concerns despite the authors’ clarifications and updates. While the authors addressed some presentation issues and provided additional explanations regarding computational overhead and theoretical results, key weaknesses remained unresolved. Reviewers were not convinced by the practical utility of the proposed framework, as no new experiments or concrete algorithmic advancements were introduced. The lack of strong experimental validation and actionable guidance continued to be major drawbacks. Additionally, the scalability of the framework to larger models and more complex multi-domain scenarios was left unsubstantiated. As a result, the reviewers upheld their initial assessments, with no significant shift in their evaluation scores.

---

### Decision · Program_Chairs · 2025-01-22

Reject